# Associations between Early-Life Food Deprivation and Risk of Frailty of Middle-Age and Elderly People: Evidence from the China Health and Retirement Longitudinal Study

**DOI:** 10.3390/nu13093066

**Published:** 2021-08-31

**Authors:** Chen Ye, Sumiya Aihemaitijiang, Ruoyu Wang, Mairepaiti Halimulati, Zhaofeng Zhang

**Affiliations:** Department of Nutrition & Food Hygiene, School of Public Health, Peking University Health Science Center, Haidian District, Beijing 100191, China; 1510306235@pku.edu.cn (C.Y.); 1410606101@pku.edu.cn (S.A.); 1710108607@bjmu.edu.cn (R.W.); 2011210145@bjmu.edu.cn (M.H.)

**Keywords:** food deprivation, frailty, childhood, old age

## Abstract

Background: The association between childhood food deprivation (FD) and health in later life has been extensively studied; however, studies on the association between childhood food deprivation and frailty are scarce. This study assessed the associations between childhood FD and the risk of frailty at middle-age and old age. Methods: Three waves of the China Health and Retirement Longitudinal Study (CHARLS), including 11,615 individuals aged over 45 years, were used for this research. Frailty was operationalized according to the FRAIL scale as a sum of fatigue, resistance, ambulation, illness, and the loss of weight. Childhood FD experiences and levels were measured by self-reported FD and historical content. Logistic mixed-effects models and proportional odds ordered logistic regression models were used to analyse the association between childhood FD and frailty. Findings: Childhood FD increased the odds of frailty at old age (1.30, 95% CI: 1.26–1.36). Compared with subjects with mild FD, those with extreme FD experiences had increased risks of frailty (1.34, 95% CI: 1.26–1.43). Subjects exposed to hunger at different ages all had an increased risk of frailty, and subjects who had FD during ages 6–12 (1.15, 95% CI: 1.09–1.22) were more likely to have an increased risk of frailty than those who had experienced FD in younger ages. The interaction of experience of FD at ages 0–6 and the experience of FD at ages 6–12 is not statistically significant after adjusting all covariates. Conclusions: Our findings suggest that childhood FD exerts long-lasting effects on frailty among older adults in China. The prevention of childhood FD may delay or even avert the emergence of frailty in people of middle-age and old age.

## 1. Introduction

With improved socio-economic conditions, more people are surviving into old age, and the prevalence of frailty is increasing and has become an urgent public health issue, especially in countries with rapidly aging populations [1,2,3]. Frailty is a clinical syndrome in geriatrics, characterized by a cumulative function decline of multiple systems and an increasing vulnerability to stressors [4,5,6]. It is associated with several adverse health outcomes, including falls, increased mortality, and catastrophic health expenditures [6,7]. Conceptual definitions of frailty include physiological, psychological, and social dimensions. Operational definitions of frailty include nutritional state, physical activity, energy, muscle strength, disability, cognitive function, and social relations and support [8].

The global prevalence of frailty is unknown yet and varies across studies because of a lack of standardisation of concepts or measures. Studies conducted in high-income countries found a weighted average estimate of 11% for frailty [4]. Previous research has shown that the prevalence rates of frailty varied between 5.9% and 17.4% in China [9]. Compared with robust subjects, frail subjects had a higher risk of all-cause mortality. The overall adjusted hazard ratio for all-cause mortality per 0.1 increment in the frailty index is 1.68 [10]. A graded increase in the risk of several causes of death was also observed among frail subjects [10]. 

Several factors, such as lower socioeconomic status and lower educational attainment, were reported to be associated with frailty [11]. As the number of frail people will increase rapidly with population aging, modifiable risk factors of frailty need to be identified. Previous research studies on the life-course epidemiology framework has illustrated how life conditions from the prenatal period impact health later in life [12]. Those who experienced childhood adversity seem to have a higher risk of cardiovascular diseases, poor cognitive function, and higher mortality rates [13,14]. Specifically, the relationship between childhood conditions and later life health may be explained by the cumulative dis/advantage theory, defined as the systemic tendency for interindividual divergences in given characteristics with the passage of time [15]. Besides, though exposures to particular environments and experiences appear to influence health development at all stages, the early-life period is thought to be important for explaining adult functioning and well-being in later life through different possible pathways [16].

Early-life FD, as one of the poor childhood conditions or childhood adversities, has been suggested as an important risk factor of negative health outcomes in adulthood. The developmental origins of health and disease (DOHaD) emphasized the importance of nutrition in intrauterine and infancy. Studies of the Ukraine famine and the Dutch “Hunger Winter” famine suggested that the exposure to FD in utero was associated with an elevated risk of such diseases as diabetes, cardiovascular disease, and hypertension in later life [17,18,19,20]. Most of these studies used famine areas and periods caused by military blockade or battle to represent FD exposure, leading the association between childhood FD and adulthood diseases to inevitably be affected by World War II [21]. China, to some extent, provides a unique background for studying the association between early-life FD and adulthood frailty, as there were several famines in China in the past century [22], which aroused much attention from scholars. 

Although previous studies have demonstrated the association between poor early-life socioeconomic conditions and frailty [23,24,25], few of them focused on the association between childhood FD and frailty. In consideration of evidence on the independent effect of early-life FD with non-communicable diseases and theories, we present the research from China to examine the effect of childhood FD and the risk of later-life frailty. Furthermore, since childhood nutrition is more likely to be improved by favorable policies, such as school nutrition, than other life-course conditions, this research might provide reference to the making of effective frailty prevention and intervention strategies. 

## 2. Materials and Methods 

### 2.1. Design and Subjects

The data were obtained from the China Health and Retirement Longitudinal Study (CHARLS), which was designed to collect a nationally representative sample of Chinese residents, aged from 45 and older, based on their economic, social and health conditions. The baseline survey of CHARLS was conducted in 28 provinces across China in 2011, and follow-up surveys were carried out every 2 years [26]. 

CHARLS used multistage stratified probability-proportionate-to-size sampling and collected high quality data via one-to-one interviews with a structured questionnaire. The data included individual weighting variables to ensure that the survey sample was nationally representative. Compared with the Chinese population census of 2010, CHARLS is quite similar to the Chinese national population [27]. A detailed description of objectives and methods of CHARLS has been reported elsewhere [26]. The Biomedical Ethics Review Committee of Peking University approved CHARLS. All participants signed written informed consent. The ethical approval number was IRB00001052-11015.

For this research, individuals who had at least 1 complete measurement of frailty from the 3 waves of surveys (Wave 2011, 2013 and 2015, respectively) and who had participated in the life history survey (2014) were chosen as the subjects (N = 12,812). To better find the association between childhood FD and frailty in middle-age and old age, subjects were limited to those that were aged between 45 and 100 years (N = 11,706). Ninety-one subjects were excluded because of the lack of covariates data. In the end, 11,615 subjects were analyzed (Figure 1).

### 2.2. Measurement of Frailty

Frailty was measured by the FRAIL scale, one of the most widely used evaluation tools of frailty [28]. The FRAIL scale in Chinese achieved semantic, idiomatic, and experiential equivalence and can apply to Chinese community-dwelling older adults [29]. It is defined as the presence of at least 3 of 5 specific attributes. These attributes are fatigue, resistance, ambulation, illness, and the loss of weight [30]. In this research, the criteria for frailty were:

Fatigue: self-report of feeling tired more than 3 days a week; 

Resistance: the question “Do you have difficulty with climbing several flights of stairs” was used and the criterion was fulfilled when subjects reported “Yes, I have difficulty and need help” or “I cannot do it”;

Ambulation: the question “Do you have difficulty with walking 1 km?” was used and the criterion was fulfilled when subjects reported “Yes, I have difficulty and need help” or “I cannot do it”;

Illness: self-report of 5 or more illnesses out of 11 illnesses;

Loss of weight: unintentional loss of 5 or more kilograms between 2 consecutive waves, based on the physical examination conducted during each survey.

The FRAIL scale scores range from 0–5 and represent frail (3–5), pre-frail (1–2), and robust (0) health status, respectively. Pre-frail and frail states were combined to (pre)frail as opposed to non-frail in analysis to create a binary outcome.

### 2.3. Measurement of Childhood Food Deprivation

Information of childhood FD was collected in the life history survey of CHARLS (2014). Moderate childhood FD was identified if respondents reported that they didn’t have enough food to eat before age 12. Extreme FD was determined by the following criteria:(1)Self-reported FD experience before age 12;(2)Born and brought up in famine-affected areas and in famine periods shown below;(3)Born and brought up in the Great Famine and had either their immediate families or siblings starved to death, with delayed conception, with the inability to conceive a child, or having had an abortion in the Great Famine.

Subjects were considered as having had an extreme FD experience if they met criteria 1 and any of the other two criterions.

Famine areas in criteria 2 refer to the Guangdong and Hunan provinces, 1946–1947 (southern famine); the Henan province, 1942–1943 (famine in central China); the Henan and Shandong provinces, 1938 (famine due to the breach of the Yellow River); the Anhui, Jiangsu and Hubei provinces, 1931 (due to the Yangtze River flood); or the Hebei, Gansu, Shaanxi and Shanxi provinces, 1928–1930 (northern famine). 

Although some famine data were not accessible because of wars and administrative problems, it can generally be said that in these chosen famines, 40–90% of farmlands had been destroyed or the agricultural production had been reduced [31]. Two thirds of residents in those areas could not get food support and approximately 50–70% of the registered population in those areas were lost (died, fled from famine, or had been trafficked) [31]. Therefore, based on the available literature, subjects growing up in these chosen areas were deemed to have experienced extreme FD.

The Great Famine in criteria 3, which happened in 1958–1962, was quite different from other famines. The scope of influence spread across China and caused more than 30 million people to die from FD, exceeding the number of people who died in World War II [32]. Although food production reduction existed nationwide, the food supply status was regulated by food rationing in the time of the Great Famine. This means that the food supply sufficiency of a province is dependent on the nationwide food stock, hence these subjects were only identified as having had extreme FD when they reported that this adverse event happened to their immediate family or siblings during the Great Famine in this research.

### 2.4. Covariates

Referring to the results of previous studies, this research included socio-demographic and health and lifestyle covariates. Socio-demographic covariates included age, gender, education level (illiterate, can read and write, primary school, junior school, high school and above), marital status (married, divorced or widowed, and never married) and the area of residence (rural and urban). Health and lifestyle covariates included body mass index (BMI), current smoking status and current drinking status (drink more than once a week). BMI was calculated as body weight divided by height squared and categorized as underweight (<18.5), normal (18.5–23.9), overweight (24.0–27.9) and obese (≥28.0) [33].

### 2.5. Statistical Analysis

Logistic mixed-effects models were applied to process the nested structure data [34]. Associations of repeated measurements of frailty and other covariates were analyzed. Additionally, this model does not require an equal number of observations from all subjects, so subjects were all included in the sample if they provided no less than 1 wave outcome. 

For each outcome, 3 models were generated: The first model was adjusted for age and gender. The second model included adjustments for age, gender, and other socio-demographic factors, such as education and marital status. The third model added all the other covariates. We also used proportional odds ordered logistic regression models, where frailty was divided into 3 categories: frail, pre-frail, and non-frail, to examine the associations between different levels and times of FD with adulthood frailty.

Finally, we performed a series of sensitivity analyses. We excluded subjects older than 85 years or younger than 60 years. In addition, by using a dichotomous outcome, we compared two groups, non-frail with pre-frail individuals and frail with other than frail individuals, to check the association in different frailty states. We also ran stratified analyses by gender, since the prevalence of frailty is expected to differ between men and women. All statistical analyses were conducted by the R (version 3.5.1 Copyright © 1998–2020, Kurt Hornik, San Diego, CA, USA) and were based on a 2-sided significance of *p* value < 0.05.

## 3. Results

### 3.1. Subjects Characteristics

The study population consists of 11,615 (53.5% female) people with a mean age of 61.6 years. The prevalence of physical pre-frailty and frailty were 38.7% and 5.0%, respectively. In men, 35.7% were pre-frail and 3.0% were frail; in women, 42.6% were pre-frail and 5.8% were frail. In general, the prevalence of frailty increased with age, and women are more likely to be frail or pre-frail than men. Subjects with lower educations and living in rural areas are more likely to be frail or pre-frail, which is consistent with previous studies in China and other countries [3,35]. The subjects’ demographic characteristics are shown in Table 1.

The prevalence of frail and pre-frail increased with increasing exposure to FD. For example, among people aged 65–74 years, 752 (28.9%) of 2610 subjects who had FD experience had frailty or pre-frailty versus 659 (23·5%) of 2809 subjects who did not have hunger experience (Figure 2).

Appendix A presents the FD experience of subjects. More than half (60%) of the subjects (63% of men and 58% of women) had experienced FD before the age of 12, whereas 1490 (12.8%) of the subjects experienced extreme FD (12.9% of men and 12.7% of women) in their childhood.

### 3.2. FD in Childhood and Risk of (pre)Frailty over Aging

The results for the association between childhood FD and the risks of being frail are shown in Figure 3. Compared to subjects who had no experience of FD, exposure to moderate FD (1.23, 95% CI: 1.18–1.28) and extreme FD (1.67, 95% CI: 1.56–1.78) could increase the risk of frailty in adulthood even after potential confounders, such as age, sex, and education were taken into consideration (Table 2). This suggests that childhood FD could raise the risk of being frail in later life. The complete results can be found in Appendix A.

We also tested whether severe FD may cause more serious frailty in people who experienced FD. Figure 4 shows the results for the association between extreme FD and the odds of being frail among subjects that reported to have experienced FD in their childhood. For those who experienced the shortage of food supply in childhood, the more serious FD was, the more they would likely be frail in adulthood (1.34, 95% CI: 1.26–1.43, *p* < 0.001). The complete results can be found in Appendix A.

### 3.3. Sensitivity Analyses

We conducted three sensitivity analyses and the results of these sensitivity analyses were in line with the findings in the main analysis. First, we excluded people aged older than 85 or younger than 45. The likelihood of having frailty was increased with older age, among women and in the higher education group, while marital status became less important (Appendix A). Then, we repeated our analyses by using frailty versus pre-frailty and non-frailty (Appendix A). Finally, we investigated the association separately by gender. The stratified analyses showed some gender differences with the main analysis (Appendix A). In general, men are more susceptible to weight and education status.

## 4. Discussion

We exploited data from CHARLS to estimate the long-term effect of the exposure to FD during childhood on the risk of frailty in adulthood. This research found that the incidence of frailty was increased in subjects with childhood FD and it remained after adjusting for a range of covariates including socio-demographics and health and lifestyle factors. We also investigated whether different degrees or ages of hunger matters. More severe famine exposure will cause more damage. Individuals who had been exposed to FD at ages 6–12 appeared to be more affected than those exposed to FD at a younger age. The interaction of hunger at different ages was not associated with the risk of frailty, showing that there is no correlation between the experiences of FD in different age groups.

Many studies investigated health outcomes in cohorts exposed to famine or early-life risk factors and evidence shows that childhood is a period in which FD is detrimental for future health. It was reported that individuals exposed to the “Dutch famine” at ages 11–14 were associated with a higher probability of developing diabetes mellitus and/or peripheral arterial disease at ages 60–76 [20]. Individuals exposed to “Leningrad suffer” in childhood and puberty may have long-term effects on systolic blood pressure and circulatory disease [18]. Compared to those who did not experience FD, individuals exposed to food deficiency during childhood face an increased risk for cognitive impairment [36]. Our findings are consistent with these results that individuals who experienced childhood hunger have a higher risk of frailty in mid and old age.

There are several potential mechanisms underlying the association of famine exposure in childhood and frailty in adulthood. The developmental origin hypothesis and epigenetic dysregulation can both explain the underlying physiological mechanism to a certain extent [37,38]. Furthermore, in periods of undernutrition, the body tends to focus on brain development by sacrificing the growth of muscles [39], and the latter is thought to be relevant to frailty. Lower muscle mass or loss of muscle mass is considered a precursor syndrome or the physical manifestation of frailty [40,41]. Exposure to FD may also increase the risk of frailty by reducing muscle in childhood. Therefore, it might be possible that even a normal lack of food can have serious consequences.

Evidence on the critical window of exposure was inconsistent. Some studies found that individuals exposed during early childhood were more strongly affected [42], whereas some identified mid- and late childhood as potential critical periods of exposure [20]. Our study found that individuals exposed after age 6 were more sensitive to frailty. The observed relationships of critical windows may be explained in several ways. Firstly, catch-up exists in the process of children’s growth and development [5]. The damage caused by malnutrition in the early stage may be partly compensated with an adequate food supply in later life. Secondly, the development of tissues and organs, such as muscles or adipose tissues, lasts until the end of adolescence, and adolescence is also a period of life during which they actively develop. These tissues may play a more important role in the occurrence of frailty. Finally, previous studies have shown that the nutritional conditions after exposure to adverse conditions also have an important influence. When the food supply was plentiful again after famine, exposed children may overeat, and sudden and large changes in nutritional status may play an important role in explaining the observed relationship [43].

We also found that different degrees of food deficiency will lead to different outcomes. Among those who had experienced FD in childhood, subjects who were exposed to extreme famine were more likely to be frail than moderate experiencers. This is consistent with similar studies of other diseases [17] and strengthens our conclusions about famine exposure and frailty risk. However, whether there would be potential dose-response patterns between food shortage and adulthood frailty is unclear since we cannot get a more individualized food supply data.

Our study has several important strengths. First, we used repeatedly measured data from a large representative sample of middle and older age Chinese adults. The sample achieved a nationally representative coverage and the data were collected through face-to-face interviews, resulting in a high response rate and high quality. Second, missing outcomes were handled by using logistic mixed-effects models. Finally, from the perspective of life processes, China provides a unique context to study the associations with FD, one of the important poor childhood conditions. The registered residence system in China can be traced back to 220 BC and is still used today, named as the Chinese Household Registration System (hukou) in present time [44,45]. People who lived in urban areas got food by a “food rationing system” (ended in 1992), while rural residents were tied to the land to produce an agricultural surplus and to deal with the volatility of agricultural production [46]. Therefore, the subjects’ life histories can be tracked and the state of food supply can be determined by the regions. Next, famous famines in Modern China took place during very well-defined time periods and areas, except for the Great Famine. Most of the time, only specific provinces were exposed to the famine whereas other provinces had adequate food supplies. 

We acknowledged there are some potential limitations. Individuals’ hunger experiences were measured by self-reported data, which may recall bias in this study. In addition, memories about childhood FD happening before 6 years old might be more vague and inaccurate than those that happened after 6 years old. Second, there are a number of tools that can be used to screen frailty, where the results of frailty can be different according to different tools [47]. We used the FRAIL scale, one of those tools which has been culturally adapted in different countries and research. Additionally, besides the lack of food itself, hunger may be accompanied by other adverse conditions and some of the hunger experiences occurred within the war. This study shows the effects of hunger but did not separate it from other contextual factors. Finally, our findings should be interpreted with caution in consideration of the potential survivor bias. If the individuals who were least healthy were more likely to die, the pool of survivors may be of better health. In that case, the average health in a population that was intensely affected by hunger could be better than the average health of a less-affected population, causing us to underestimate the impact of exposure to hunger on health.

## 5. Conclusions

In recent years it has become increasingly recognized that it is very important to identify the early factors affecting the health of the growing elderly population. Our study shows significant associations of exposure to FD at childhood with frailty. Exposure to undernutrition at ages 6–12 is associated with a higher probability of developing frailty in later life. However, whether there is a potential dose-response pattern remains to be studied in the future. In this case, it is reasonable and possible for policy makers to have more effective preventive strategies and interventions to health development by increasing investments in childhood prevention or school nutrition programs and expand the concept of frailty beyond diseases and disability.

## Figures and Tables

**Figure 1 nutrients-13-03066-f001:**
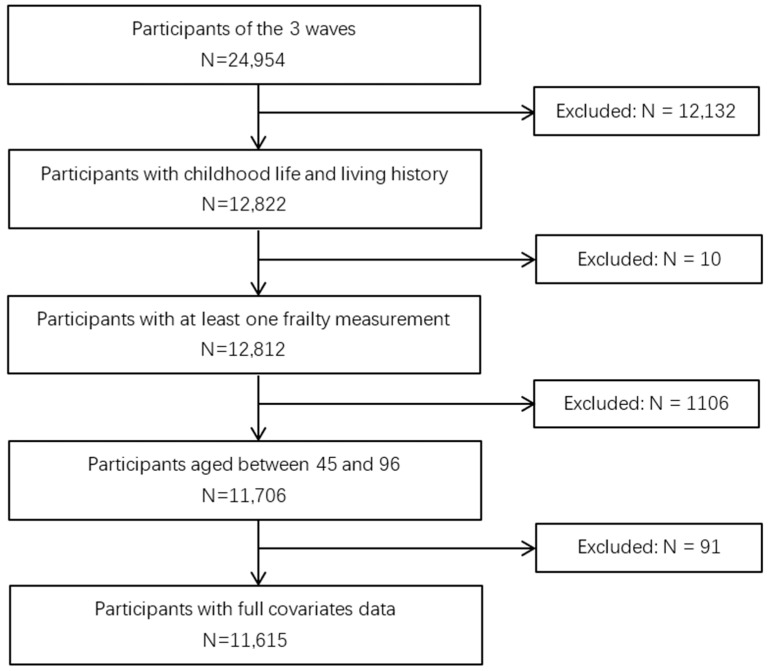
Flowchart on the sample selection and exclusion.

**Figure 2 nutrients-13-03066-f002:**
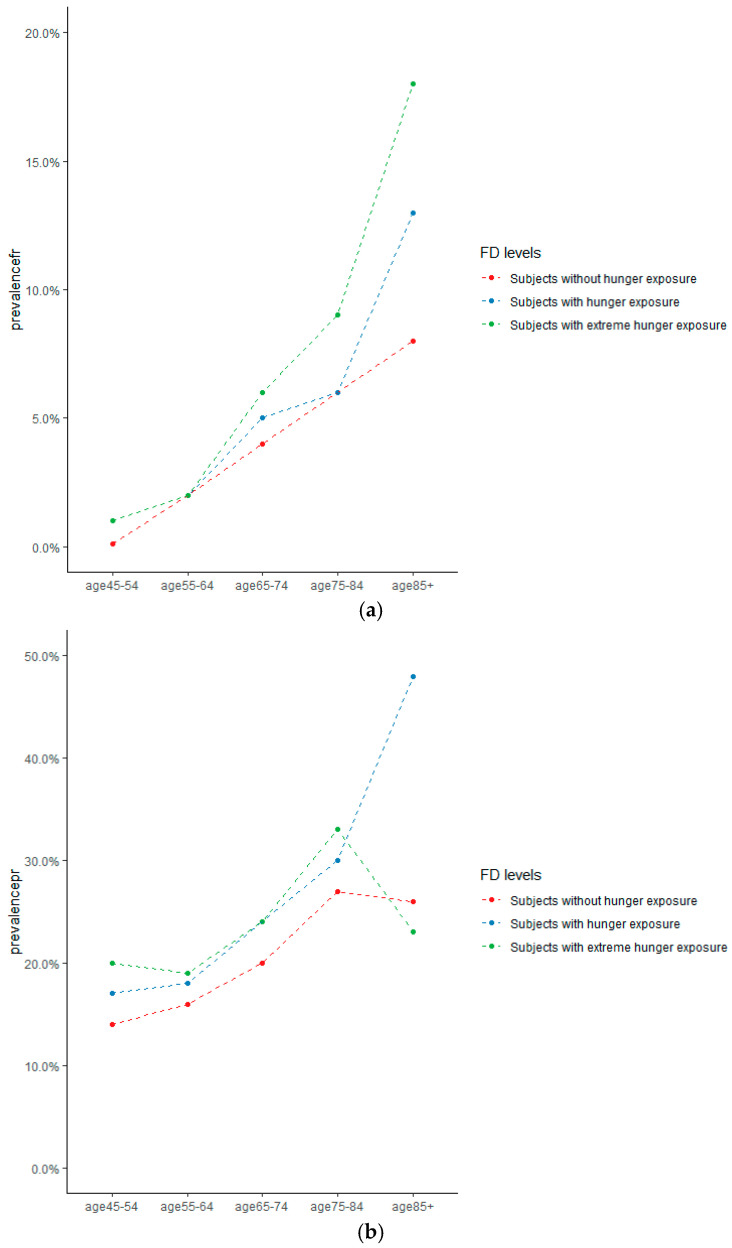
(**a**) Prevalence of frailty in China, by age and hunger exposure; (**b**) Prevalence of pre-frailty in China, by age and hunger exposure.

**Figure 3 nutrients-13-03066-f003:**
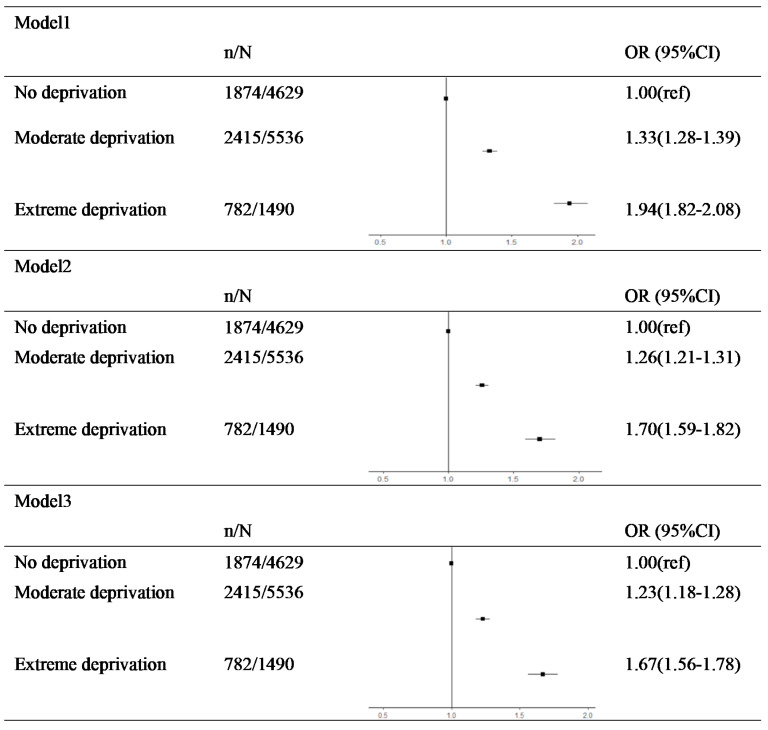
Association between FD exposure and (pre)frailty.

**Figure 4 nutrients-13-03066-f004:**
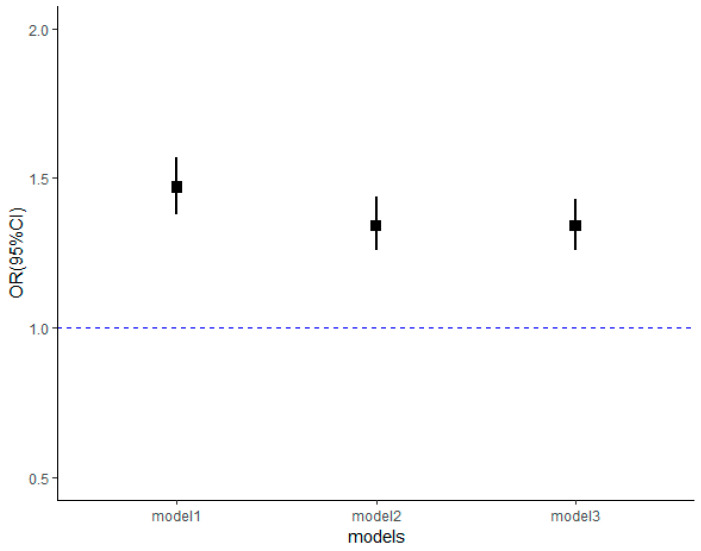
Association between extreme food deprivation and physical (pre)frailty.

**Table 1 nutrients-13-03066-t001:** Characteristics of the subjects.

Variable	Overall 11,615	Non-Frailty 6544	Pre-Frailty 4492	Frailty 579
Age, years (SD)	61.6 (9.6)	60.3 (9.3)	62.6 (9.8)	67.2 (9.3)
Gender				
Male	5388 (46.39%)	3241 (49.53%)	1872 (41.67%)	175 (3.9%)
Female	6227 (53.61%)	3203 (48.95%)	2620 (58.33%)	404 (8.99%)
Education				
Illiterate	3234 (27.84%)	1465 (22.39%)	1487 (33.1%)	282 (48.7%)
Can read and write	2265 (19.5%)	1152 (17.6%)	994 (22.13%)	119 (20.55%)
Primary school	2636 (22.69%)	1517 (23.18%)	10013 (222.91%)	106 (18.31%)
Junior middle school	2307 (19.86%)	1535 (23.46%)	723 (16.1%)	49 (8.46%)
High school and above	1173 (10.1%)	875 (13.37%)	275 (6.12%)	23 (3.97%)
Marital status				
Married	9553 (82.25%)	5530 (84.5%)	3590 (79.92%)	433 (74.78%)
Divorced	593 (5.11%)	330 (5.04%)	244 (5.43%)	19 (3.28%)
Widowed	1389 (11.96%)	647 (9.89%)	618 (13.76%)	124 (21.42%)
Never married	80 (0.69%)	37 (0.57%)	40 (0.89%)	3 (0.52%)
Residence				
Urban	4102 (35.32%)	2584 (39.49%)	1348 (30.01%)	170 (29.36%)
Rural	7513 (64.68%)	3960 (60.51%)	3144 (69.99%)	409 (70.64%)
BMI ^1^				
<18.5	747 (6.43%)	355 (5.42%)	332 (7.39%)	60 (10.36%)
18.5–24	5843 (50.31%)	3383 (51.7%)	2242 (49.91%)	218 (37.65%)
24–28	3612 (31.1%)	2100 (32.09%)	1327 (29.54%)	185 (31.95%)
≥28	1413 (12.17%)	706 (10.79%)	591 (13.16%)	116 (20.03%)
Smoking	5015 (43.18%)	2935 (44.85%)	1871 (41.65%)	209 (36.1%)
Drinking	2987 (25.72%)	1944 (29.71%)	977 (21.75%)	66 (11.4%)

^1^ (BMI: body mass index).

**Table 2 nutrients-13-03066-t002:** Association of hunger at different age groups with (pre)frailty.

Age (Years)	Model 1 OR (95% CI)	Model 2 OR (95% CI)	Model 3 OR (95% CI)
Age 0–6	1.14 (1.06–1.18) *	1.12 (1.06–1.18) *	1.12 (1.06–1.18) *
Age 6–12	1.17 (1.14–1.21) **	1.16 (1.12–1.20) **	1.15 (1.09–1.22) **
Age 0–6 * age 6–12	1.10 (1.03–1.17)	1.09 (1.02–1.16)	1.09 (1.02–1.18)

** *p* < 0.01; * *p* < 0.05.

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
