# Peer review of "Associations between Early-Life Food Deprivation and Risk of Frailty of Middle-Age and Elderly People: Evidence from the China Health and Retirement Longitudinal Study"

_nutrients, 2021, doi:10.3390/nu13093066_

Round 1

Reviewer 1 Report

In this study the authors sought to explore the association between childhood food deprivation (FD) and frailty by, specifically, assessing the associations between childhood FD and the risk of frailty at middle-age and old age. For this, 3 "waves" (2011, 2013 and 2015), of the China Health and Retirement Longitudinal Study (CHARLS), which comprises  over 11,000 individuals (>45 yo) representative of the Chinese resident population, were used. Methodologically, frailty was operationalized according to the FRAIL scale (sum of fatigue, resistance, ambulation, illness, and the loss of weight), while childhood FD experiences and levels were measured by self-reported and historical content. Overall, it is concluded that childhood FD increases the odds of frailty at middle/old(er) age.

Given the sample size, and overall historical context, this paper is actually a relevant addition to the literature. These type of studies have always inherent limitations (can FD be separated from other contextual variables, for instance), but nonetheless the authors were careful in their analysis, acknowledge shortcomings, and are careful in their interpretations of the findings.

Below a few notes:

ABSTRACT

Overall, it is well written and summarizes well the study, not over-reaching on conclusions.

INTRODUCTION

1.The first sentence reads "strangely". It appears that perhaps the authors are attempting to "put too much" into the first sentence. I would suggest dividing the sentence into two separate sentences: one focusing on we are living longer, a second on the need for studies on frailty/determing factors. There might be a further oversight as the references start with #6-8;

2. The exact same first paragraph is is repeated as the second  paragraph. This might just have been an oversight by the authors, put should be verified and corrected for;

3. Please review sentence in line 60, the verb "taken" does not appear to be the more correct one for the sentence (using, perhaps?). In line 70 the present perfect may be more appropriate that the past perfect (they had been, but are not anymore? if still is, "have been" is a better option. I am not sure what the authors mean by the opening "in consideration of researchers" on line 84, "in line with" perhaps?  There are other slight grammar/word inconsistencies, but a careful reading by the authors (the introduction is overall very well writen) should resolve any issues;

METHODS

1.There is an error on line 102, n=11,706 (not 1,106). On line 129 "childhood" should start with capital "C". Line 144 should read "in the chosen famines". An additional reading by the authors, should resolve these small issues;

2. Please indicate if the FRAIL scale has been previously validated for the Chinese population, and/or briefly explain why this scale was chosen for the study;

3. Briefly explain inclusion of why 45 yo (or thereabouts) in the inclusion criteria. For these individuals, birth year (about 1966 at wave 1 of asssessment) correspondents to outside a famine period. Is this so to include time periods inside/outside famine/more recent time periods?;

4. Not clear in Line 178, why the inclusion criteria for age was further narrowed down in further (sensitivity) analysis. Does it relate to the famine historical periods? Wouldn't the results be "richer" if including all? This does not become clearer after reading lines 231/232, thus needs some further context/explanation;

RESULTS

1. It is not cleat what you mean in sentence 202/203. Is this particular time period selected for a table/figure? Selected for what?;

2.The resoltuion of Fig.2 is poor. Legends and axis are very difficult to read, and not clear what is meant by "prevalencefr" (I assume it is frailty?) vs "prevalencepr" (pre-frailty?). Perhaps spell it all out? There is also not need to repeat "age" before each age group in the X axis, the axis legend already reads "age";

CONCLUSIONS/DISCUSSION

1. Again, a few grammatical issues. For example, in line 252 "Individuals experienced Leningrad suffered" needs revising, in line 256 where it reads "hungry" should be "hunger", line 264 should read  "it might be possible", line 265 "Evidences on the critical window of exposure have been inconsistent", and line 281 "subjects who were exposed", line 305-306 needs some revising ("lots of" is a rather "open term", perhaps "a variety of", and previously you referred to the FRAIL scale and not to the "frail scale"...). This reviewer advises for a throughout reading by someone external to the paper (sometimes from the outside it is easier to catch these incongruencies);

2. In line 263 can you go a bit deeper than "and the latter is thought to have a lot to do with frailty". A lot to do in what sense?

Author Response

Response to Reviewer 1 Comments

Point 1: The first sentence reads "strangely". It appears that perhaps the authors are attempting to "put too much" into the first sentence. I would suggest dividing the sentence into two separate sentences: one focusing on we are living longer, a second on the need for studies on frailty/determing factors. There might be a further oversight as the references start with #6-8;

Response 1: Sorry for my mistake. We checked the references and rewrote the sentence.

(line 34-36)

Point 2: The exact same first paragraph is repeated as the second paragraph. This might just have been an oversight by the authors, put should be verified and corrected for;

Response 2: Sorry for my mistake. It was an oversight and we corrected the paragraph.

(line 43)

Point 3: Please review sentence in line 60, the verb "taken" does not appear to be the more correct one for the sentence (using, perhaps?). In line 70 the present perfect may be more appropriate that the past perfect (they had been, but are not anymore? if still is, "have been" is a better option. I am not sure what the authors mean by the opening "in consideration of researchers" on line 84, "in line with" perhaps?  There are other slight grammar/word inconsistencies, but a careful reading by the authors (the introduction is overall very well writen) should resolve any issues;

Response 3: Thank you for your suggestion. We changed the statement at your suggestion. We reorganized the language to express our research was based on evidences of early life FD with non-communicable diseases and theories of early life.

(line 52, 62, 75)

Point 4: There is an error on line 102, n=11,706 (not 1,106). On line 129 "childhood" should start with capital "C". Line 144 should read "in the chosen famines". An additional reading by the authors, should resolve these small issues;

Response 4: Sorry for my mistake. We corrected the typing errors.

(line 96, 122, 135)

Point 5: Please indicate if the FRAIL scale has been previously validated for the Chinese population, and/or briefly explain why this scale was chosen for the study;

Response 5: Thank you for your suggestion. The FRAIL scale has been validated for Chinese and the Chinese version is available. Studies on the reliability and validity of using the FRAIL scale to evaluate frailty of Chinese population showed good results. We added evidence of application of the FRAIL Scale for Chinese population.

(line 103-104)

Point 6: Briefly explain inclusion of why 45 yo (or thereabouts) in the inclusion criteria.  For these individuals, birth year (about 1966 at wave 1 of asssessment) correspondents to outside a famine period.  Is this so to include time periods inside/outside famine/more recent time periods?

Response 6: Sorry for my unclearly express. We hoped to see the effect in middle-aged and older people. Therefore, we included individuals over 45. Individuals born or grown after about 1965 were outside famine period. These individuals wouldn’t be judged as “extreme FD experiencer” for they haven’t exposed to famines. But they can be judged as “moderate FD experiencer” if they reported that they didn’t have enough food to eat before age 12.

In this study, childhood FD was identified by self- report hunger experience and exposure to famines only affects the determination of hunger. For example, if an individual reported hunger experience (criteria 1) in the article, line 125), and we found that this hunger period happened in famine period (criteria 2) or criteria 3) in the article, line 126 and 127), this individual was identified as “extreme FD experiencer”. Meanwhile, if an individual reported hunger experience, and we found that this hunger period was not happened in famine period, then this individual was identified as “moderate FD experiencer”. By the way, if an individual grown in famine period and famine area but reported no hunger experience, this individual was not considered to have food deprivation. Famine periods won’t influence judgment of food deprivation. Therefore, it is ok to included time periods inside/outside famine/more recent time periods.

(line 124)

Point 7: Not clear in Line 178, why the inclusion criteria for age was further narrowed down in further (sensitivity) analysis. Does it relate to the famine historical periods? Wouldn't the results be "richer" if including all? This does not become clearer after reading lines 231/232, thus needs some further context/explanation

Response 7: Sorry for my unclearly express. No, it does not relate to famine periods. We performed this sensitivity analysis in order to observe this association in old age individuals. Besides, individuals over 85 years old have greater survivor bias than others. In order to eliminate the interference of these unknown factors we excluded subjects older than 85 years or younger than 60 years in sensitivity analysis.

(line 168)

Point 8: It is not cleat what you mean in sentence 202/203. Is this particular time period selected for a table/figure? Selected for what?

Response 8: I‘m so sorry because I didn’t make it clear. We verified the accuracy of the selected famine period by calculating the rate of reported hunger experience rate in these famine periods. The result showed that the FD rate (number of individuals reported hunger experience/total number) in famine periods are higher than significantly higher than the general average. But we found that this had little to do with the result of the article, so we decided to delete this sentence.

(line 88)

Point 9: The resoltuion of Fig.2 is poor. Legends and axis are very difficult to read, and not clear what is meant by "prevalencefr" (I assume it is frailty?) vs "prevalencepr" (pre-frailty?). Perhaps spell it all out? There is also not need to repeat "age" before each age group in the X axis, the axis legend already reads "age";

Response 9: According to your suggestion, we redrawn Fig 2. Fig 2a. represent prevalence of frailty and Fig 2b. represent prevalence of pre-frailty. We can submit PDF images to avoid low resolution 

(line 191)

Point 10: Again, a few grammatical issues. For example, in line 252 "Individuals experienced Leningrad suffered" needs revising, in line 256 where it reads "hungry" should be "hunger", line 264 should read "it might be possible", line 265 "Evidences on the critical window of exposure have been inconsistent", and line 281 "subjects who were exposed", line 305-306 needs some revising ("lots of" is a rather "open term", perhaps "a variety of", and previously you referred to the FRAIL scale and not to the "frail scale"...). This reviewer advises for a throughout reading by someone external to the paper (sometimes from the outside it is easier to catch these incongruencies);

Response 10: I apologize for my mistakes. We corrected grammar and spelling mistakes. “Leningrad suffer” happened in Russia and it is also one of the research hotspots. There are many tools to screen frailty (Fried scale, FRAIL scale, Tilburg Frailty Index, et al). Lots of in line 283 refer to these screening tools. We replaced the “lots of” to avoid ambiguity.

(line 234, 238, 247, 261, 283)

Point 11: In line 263 can you go a bit deeper than "and the latter is thought to have a lot to do with frailty". A lot to do in what sense?

Response 10: Sorry for my unclearly express. “a lot to do” in this sentence means “relevant to”. We replaced it to avoid ambiguity and further elaborated the underlying mechanism.

(line 243-246)

Reviewer 2 Report

This review covers a manuscript entitled " study on the association between childhood FD and frailty of adulthood" by authored by Chen Ye et al. 

  1. It is necessary to describe the sampling method

And because the subjects are people drawn from a specific area and at a specific time of famine.

It cannot be said to be a representative sample of middle and older Chinese adults.

  1. When researchers collect face-to-face data, It was not stated whether ethical concerns procedures were followed.
  2. Adulthood heath status and frailty by early-life food deprivation, It has been mentioned in numerous papers in the past. This paper investigated the frailty of middle and old age of young child who suffered a great famine. This is because the research results are sufficiently predictable based on several existing papers. Compared to the great effort for research, scientific interest, specificity and novelty of the content have decreased.

Author Response

Response to Reviewer 2 Comments

Point 1: It is necessary to describe the sampling method

And because the subjects are people drawn from a specific area and at a specific time of famine.

It cannot be said to be a representative sample of middle and older Chinese adults.

Response 1: Sorry for my unclearly express. The entire population came from CHARLS(a representative sample of middle and older Chinese adults, not just form specific area) and all of the subjects were included in the analysis. Only people with self-reported hunger experience from specific area and at a specific time of famine were recognized as “extreme food deprivation experiencers”. People not from specific area or famine period were recognized as “moderate food deprivation experiencers” or “individuals without food deprivation experience”. Therefore., it was a representative sample.

(line 83, 122)

Point 2: When researchers collect face-to-face data, it was not stated whether ethical concerns procedures were followed.

Response 2: Thank you for your suggestion. We added ethical procedures description. All participants signed written informed consent in CHARLS.

(line 89-90)

Point 3: Adulthood heath status and frailty by early-life food deprivation, It has been mentioned in numerous papers in the past. This paper investigated the frailty of middle and old age of young child who suffered a great famine. This is because the research results are sufficiently predictable based on several existing papers. Compared to the great effort for research, scientific interest, specificity and novelty of the content have decreased.

Response 3: Thank you for your suggestion. Most of previous papers looked at childhood adversity with frailty, not food deprivation. Early- life food deprivation was often considered one of the types of childhood adversity, but few studies have independently analyzed the association between food deprivation and frailty. We focused on food deprivation rather than the effects of childhood adversity. Unlike other kind of childhood adversity, food availability can easily be improved through policies, for example, “school nutrition policies”. Furthermore, we analyzed the effects of different hunger levels and periods of hunger and found that there was a time window for the effects of childhood food deprivation on frailty. These are not available in previous researches.

Round 2

Reviewer 2 Report

I reviewed your revised paper.
The reliability of the IRB number presented in the revised paper
must be confirmed before the start of the study, and if confirmed,

Author Response

Response to Reviewer 2 Comments

Point 1: China is one of the most populous countries in the world, and the explanation of the method of extracting subjects that can represent them was insufficient to understand many people

Response 1: Thank you for your suggestion. We supplement the sampling method and information of CHARLS.

Point 2: The IRB approval period and proof of the start of the research must be specified in the paper to be recognized for its reliability.

Response 2: Thank you for your suggestion. I apologize for my earlier oversight. The Biomedical Ethics Review Committee of Peking University approved CHARLS. The research was conducted after ethical review. We supplemented the documentation of the IRB approval.

The original document is in Chinese. We simply translate it into English for reading. I apologize for any inaccurate translation.

Peking University Institutional Review Board (PKU IRB)

IRB approval

Ethical approval number: IRB00001052-11015

Project name: China Health and Retirement Longitudinal Study

Project leader: Yaohui Zhao  Title: professor      Contacts: Yisong Hu   Telephone:62766045

Main research institute: National School of Development at Peking University

Research project source: â–¡government â–¡Foundation â–¡company â–¡international organization â–¡independent â–¡others

Research funder:

Request review type: □new application ☑revised project □continuing review topics

Review comments:

After reviewed by the ethics committee, the research protocol and informed consent of China Health and Retirement Longitudinal Study:

☑ meet the ethical requirements and agree to conduct the experiment according to this scheme

â–¡after modify â–¡experimental scheme â–¡informed consent, the ethics committee agreed to start the trial

â–¡not ethical or â–¡after added information and modify and report to the Ethics Committee for review once again

â–¡terminate â–¡suspend approved experiments
